# An Integrated Environment for Monitoring and Documenting Quality in Map Composition Utilizing Cadastral Data

**Ioannis Kavadas [1] and Lysandros Tsoulos [2,*]**

[1]  Internal Quality Unit, Hellenic Cadastre, Holargos, 15562 Athens, Greece; ikavadas@ktimatologio.gr
[2]  Cartography Laboratory, School of Rural, Surveying and Geoinformatics Engineering, National Technical University of Athens, 15780 Zografou, Greece
*  Correspondence: lysandro@central.ntua.gr

**Abstract:** Topographic maps show both physical and artificial entities of the surface of the Earth which represent distinct features forming the building blocks in map composition. Their portrayal on the map is subject to constraints dependent on the method of data collection, the map scale, the data processing procedures and the requirements of map users. In addition to constraints, geospatial data contain uncertainties and errors that are either inherent in the data or a result of the map composition process. The type and significance of these errors determine the quality of maps. This paper elaborates on the development of an integrated environment for monitoring and documenting quality in the map composition process. In this environment, quality plays a vital role in all phases of map production whereby it is continuously assessed and documented. The methodology described involves the design and implementation of a "quality model" based on international Standards. An integrated software application for the utilization of cadastral information to produce and update topographic maps at a scale of 1:25,000 was also developed. The aim is to implement the proposed methodology in a real production environment and to use it as a proof of concept.

**Keywords:** data quality; quality model; international standards; 1:25,000 scale topographic map; inspire

## 1. Introduction

In geospatial data, quality is a multidimensional characteristic that is involved in different ways in all phases of their lifespan, from the collection of primary information to the creation of the final product. Although it is commonly accepted that "quality" has a significant impact on the utilization of geospatial data, it constitutes a concern for the scientific community in geoinformatics.

In the literature, two basic concepts are presented depending on the perspective of quality. According to the first concept, a "quality product" is one that fully complies with the requirements of the specifications or is free from errors, and reflects mainly the perspective of producers of geospatial data [1–3]. The term "quality" from the point of view of the geospatial data producer can be defined as "the measure of the difference between the digital data and the geographical reality they represent". It decreases as this difference increases [4]. The other view focuses mainly on the suitability of geospatial data for a specific use as it emerges from the expectations of the data user [5,6]. From the point of view of the data user, quality is more focused on their suitability for the application or use for which they are intended, in other words it "is a measure of suitability for the use of data for a specific purpose" [4].

The internationally accepted definition of quality is contained in the vocabulary of terms of the ISO 9000 family, where the term "quality" is defined as: "the degree to which a set of inherent characteristics of an object fulfills requirements" [7]. This definition adequately covers both the producer's and the user's perspective. The requirements set

by the producer relate to the requirements of the product specifications on the basis of which the product is created. On the other hand, the requirements set by the user refer to the characteristics that the dataset must possess in order to make it suitable for the expected use. A more detailed definition of geospatial data quality, which captures the above, is contained in the now-defunct ISO 19113:2002 [8] that has been incorporated into ISO 19157:2013 [1], which adopted the definition of ISO 9000. Geospatial data quality can also be defined as "fitness for use", including both quality of design, conformance to the design (production-oriented quality), customer satisfaction, and the fulfillment of the needs of the society or the environment [9].

In map composition, cartographers use different sources and various types of geospatial data usually collected for other purposes. These data are then incorporated into their applications and processed through transformations that are additional sources of error for the resulting product. It is important for cartographers, who exploit and integrate geospatial data in the map production line, to adopt quality management as a key factor to produce reliable maps.

In order to achieve the above-mentioned objective, the cartographer, drawing on international experience, should implement a quality policy and adopt a quality management system (QMS)/project management system (PMS) as an integral part of the map composition/production process [10]. The adoption of internationally recognized standards such as ISO 9001:2015 [11], ISO 10005:2018 [12], ISO 10006:2017 [13] or PMBOK [14] is considered best practice. By applying a QMS/PMS to map production, quality management is involved in all production phases, from the identification of the user's requirements to the delivery of the final product (Figure 1).

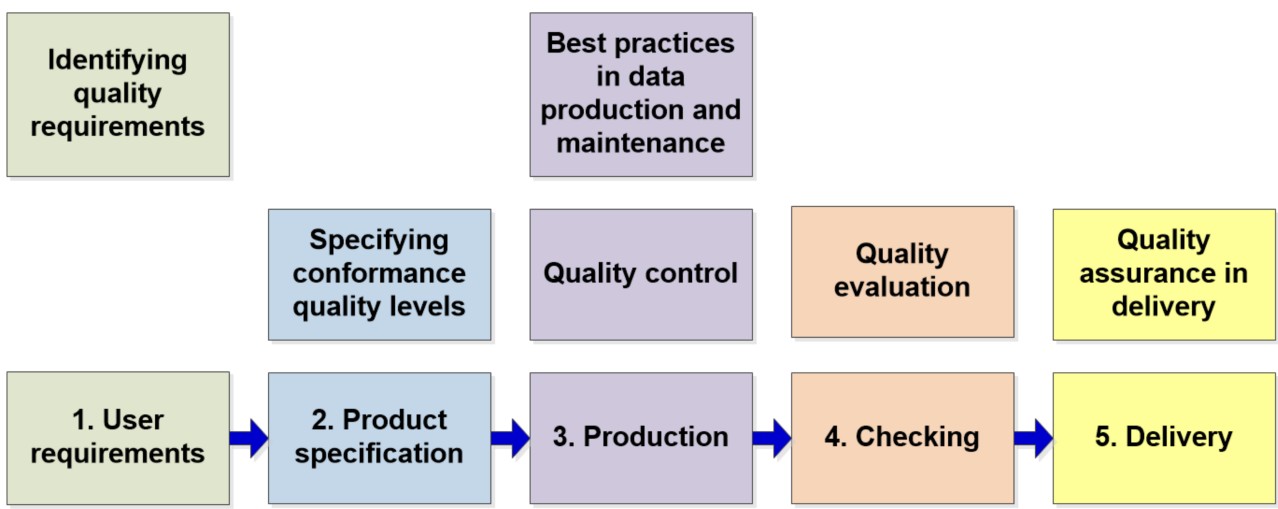

**Figure 1.** Quality in general production process [15].

Table 1 indicates the way quality management is involved in the three main phases of the geospatial data production process, namely, "before production", "production" and "after production". It is now clear that in order for the cartographer to be able to evaluate map quality, it must be adequately assessed and documented. The best way to achieve this is to make quality management part of the production process. Interpreting the contents of Table 1, the core of the quality system to be implemented is the design and implementation of a quality model.

**Table 1.** Interpretation of quality in different phases of production [15].

| Phase | Quality Documentation | Goal for Quality | Quality Methods | Level |
|---|---|---|---|---|
| **Before production** | Specification Quality model | Define quality requirements | Analysis of customer requirements | Entity/Feature type level |
| **Production** | Database Process history | Meet the specifications Record expected quality to database | Inspection | Entity/Feature instance |
| **After production** | Metadata Test reports | Measure conformance to quality requirements | Evaluation Reporting | Dataset level |

## 2. Quality Model (QM)

A definition of the quality model for geospatial data is given in [16]. It is defined as "a model describing the quality of a spatial data set according to the technical specifications" (Fit for purpose QM). According to [15] a quality model for geospatial data handles the differences between the dataset and the "Universe of Discourse" (UoD) [17] and identifies how these differences can be characterized, defined, measured and documented as metadata for spatial data. A quality model also includes organizational issues related to data quality management, e.g., the management of the differences between the dataset and the UoD. Based on the above, a geospatial data quality model is defined as "a conceptual framework for measuring and representing the quality of a dataset". A more comprehensive definition is proposed by the Quality Knowledge Exchange Network (Q-KEN) Committee of Eurogeographics, which defines a geospatial data quality model as "A framework for defining, evaluating, documenting, and presenting the quality of spatial data sets and geo-services according to their specifications" [18]. Complementing the above definition, the quality model, in addition to an integrated quality management framework, is a "tool" for standardized and systematic actions that continuously improve the quality of products and services.

The goal of successfully implementing a quality model is to ensure that the needs of map users are met in a timely and effective manner. When a QM is implemented, it provides (a) a common understanding of data quality issues across all stakeholders, (b) improved performance, (c) lower production costs, (d) confidence in the data and (e) more effective data quality management and monitoring. In geospatial data, quality information is usually referenced at the entity level. The quality model shall identify the quality requirements at the entity level, detect sources of potential errors that affect the quality of the data and define the metrics required to assess and ensure their quality.

In the context of this work, as best practice, guidelines are provided concerning the design and development of a quality model, based on the international standard ISO 19157:2013 [1] for quality monitoring in map production. The same principles can also be used to develop and implement a quality model for any product derived from geospatial data.

The effective implementation of a quality model includes the following components:

i. Design of the quality model: The QM consists of several discrete modules. The depth to which each module will be applied depends on the user's needs, the characteristics of the geospatial data under assessment and the importance of the product or service for the data/map producer [18]. The methodological approach includes: (a) a study of user requirements and identification of quality requirements, (b) a selection of quality elements using data quality (DQ) elements of the ISO19157:2013 [1], (c) an identification of quality measures using the standardized DQ measures of Annex D of ISO 19157:2013 [1] and (d) an identification of evaluation methods. In addition, the quality model may include the required levels of quality compliance (achievement of quality objectives), details of any additional

      methods used to control quality in the production flow line, definition of test plans, and instructions for the production of metadata.

ii.    Evaluation/assessment of quality by applying the quality model: The map data are tested using the evaluation methods defined in the quality model. To produce a quality result for each quality predefined measure, the flow of the process for the evaluation of data quality of ISO 19157:2013 [1] is used.

iii.    Quality Model improvement: The quality results obtained from the map data evaluation are compared to the quality objectives set by the map producer. Failure to achieve any of the quality objectives leads to a repetition of the design. Furthermore, the implementation of the QM may result in: (a) an update of the QM in terms of identifying new quality requirements, quality measures and more effective evaluation methods based on the "knowledge" gained from its implementation, and (b) the results of its implementation may also result in an update of the project management procedures.

The above-mentioned components are interdependent, and the results produced by each one of them have an effect on the other. To make the quality model "stable", it is necessary to test it on a subset of geospatial data.

In order to properly design a QM, certain basic conditions should be set, which should be met so that it can be used effectively [10]. One of the most important conditions for its design is the use of international standards and general concepts which have already been formulated. The application of quality standards in the development of a quality model requires a full understanding of certain factors, such as the nature of the domain from which the data originate, the manner of their representation in a digital environment, organization and evaluation, etc. [19].

### 3. Geospatial Standards

Standards related to geospatial information have been developed by a number of international and non-international organizations (ANSI, ASPRS, FGDC, IEC, IHO, ISO, OGC, etc.) and are divided into different categories and levels depending on the scope they refer to. Like all standards, geospatial data standards include guidelines for their effective implementation and in no case constitute or include product specifications. A standard is a conceptual/abstract description, which in order to be practically applied requires further elaboration on the manner of its application. Generally, geospatial data standards are open to interpretation as to how they are to be applied and implemented at a practical level.

The most important factors to be considered in the identification of the appropriate standards are: (a) meeting the requirements of the producer and the user, (b) the cost/benefit ratio of their implementation and (c) the interdependence between them. Since geospatial information is complex and diverse in terms of its content and management, to address any problems in understanding how to use standards, standardization organizations provide a series of standards that are mutually supportive, e.g., the ISO 19000 series. Table 2 lists the phases where the geospatial standards can be applied in the different stages of the map production process. Table 3 is an example of the application of the international standards of the ISO 19000 family.

**Table 2.** Interpretation of geospatial standards in different phases of production.

| Phase | Scope of Geospatial Standards |
|---|---|
| **Before production** | User requirements (product specifications) <br> Define quality requirements (product specifications) <br> Methods and rules of data collection |

**Table 2.** *Cont.*

| Phase | Scope of Geospatial Standards |
|---|---|
| **Production** | Compliance with specifications<br>Best practices in data/map production<br>Methods and tools of analysis and processing<br>Quality evaluation |
| **After production** | Quality assurance<br>Metadata<br>Presentation of data<br>Data access services<br>Data portability and interoperability<br>Quality improvement<br>Knowledge transfer |

**Table 3.** Application of the ISO 19000 family of geospatial standards to quality management in the production process.

| Phase | Scope of Standards | Standard | Implementation |
|---|---|---|---|
| **Before production** | Product specifications<br>Product specifications<br>Define quality requirements | ISO 19131<br>ISO 19157<br>ISO 19157 | Guidelines for creating specifications<br>Guidelines for determining compliance levels<br>Quality model design and documentation |
| **Production** | Quality elements | ISO 19157 | Selection of applicable quality elements |
| | Quality measures | ISO 19157<br>ISO 19115-1<br>ISO 19115-2 | Selection of quality measures<br>Quantitative measures according to ISO 19157<br>Non-quantitative measures according to ISO 19115-1/19115-2 |
| | Quality evaluation methods | ISO 19157<br>ISO 2859-1<br>SISO 3951-1 | Quality evaluation according to ISO 19157<br>Sampling methods according to the series of standards ISO 2859 or ISO 3951 depending on the type of quality element under evaluation |
| | Reporting metadata | ISO 19157<br>ISO 19115-1<br>ISO 19115-3 | ISO 19157 specifies the data to be recorded in the metadata and quality reports<br>ISO 19115-1/19115-2 specify the format required to describe geographic information and services through metadata |
| **After production** | Quality assurance | ISO 19158 | Measuring compliance with quality requirements |
| | Reporting | ISO 19115-1<br>ISO 19115-3 | Metadata<br>Quality reports |
| | Metadata exchange | ISO 19139 | XML schema implementation |

The standard that covers almost all the quality parameters in a satisfactory way, is the ISO 19157:2013 [1]. Adopted by FGDC [20] and OGC [21], it can be used effectively as a reference for the definition of quality elements and their evaluation as well as for the assessment/recording the quality of a geospatial dataset/map through audit reports and metadata.

## 4. Quality Monitoring in Map Composition

In map composition, geospatial data originate from: (a) a larger-scale map through generalization, (b) data collected for the purpose of map composition and (c) data collected for other purposes. In each case, the cartographer is required to evaluate, quantify and document the quality of the produced map. In case the map to be produced is derived from the use of geospatial data collected for other purposes, the cartographer is required to evaluate their quality and to assess their suitability for map composition. In the context of the re-

search for the design, analysis, documentation and implementation of an integrated quality monitoring environment for map composition, the proposed methodology is applied to the composition of a topographic map at a scale of 1:25,000. Cadastral information collected and maintained by the Hellenic Cadastre was selected as the basic geospatial information and is considered as meeting the quality requirements for use in map composition.

In the map composition process, the workflow involves three main steps: (a) the creation and updating of a geospatial database, (b) the transition and generalization of the entities of the geospatial database and the creation of a cartographic database that will include only the features to be portrayed on the map and (c) the composition of the map. In order for the cartographer to be able to monitor map composition, it is proposed to adopt and implement the following three (3) quality models designed based on ISO 19157:2013 [1]:

i. Geospatial data quality model: The QM implementation results to quantified quality information for the geospatial entities to be included in the geospatial database. The main objectives are: (a) to confirm the suitability of the data selected for map composition and (b) to obtain exact knowledge of the inherent and residual errors at the entity level due to processing.

ii. Cartographic features quality model: The QM implementation results in quantified quality information for the geospatial entities to be included in the cartographic features database. The main objectives are: (a) the exact knowledge of the errors at the feature level due to generalization and transition and (b) to detect if the transition process did not create new errors in the dataset.

iii. Quality model of the composed map: The QM implementation results in quantified quality information for the map. The aim is to provide users with evidence of control and inspection of the map, while giving, through metadata, information related to the quality of the final product.

## 5. Software Application for the Map Composition Process

The 1:25,000-scale topographic map is an accurate and detailed representation of the cartographic features at the particular scale, including the names of the main sites, administrative boundaries, etc. The 1:25,000 scale map is mainly used as reference material (base map) for the composition of other maps used in the planning of development projects, forest management, the production of maps at smaller scales as well as the production of thematic maps.

As part of this work, an integrated software application was developed for the integration of cadastral information at scales 1:1000 and 1:5000 provided by the Hellenic Cadastre to produce the topographic map at scale 1:25,000. A summary workflow of the process is shown in Figure 2.

The geospatial data of the Hellenic Cadastre were selected as the main reference information for the composition of the map. The main advantages of the geospatial information of the Hellenic Cadastre are: (a) it covers the whole country, (b) the information is available in digital form, (c) it is updated continuously in real time, (d) it is referenced to a single geodetic reference system, (e) its geometric accuracy exceeds the accuracy of the map and (f)it has appropriate and documented quality at the entity level. The software application enables the user to compose the map by selecting the area of interest. This functionality accelerates the production processes and allows the producer to better monitor and manage its quality.

The software application was developed in the Microsoft Visual Studio environment and is a stand-alone desktop application for the Microsoft Windows operating system. Visual Studio is the main management platform of the application and is used in the development of its user interface, while the basic code is developed in the Python programming language using the ArcPy site package. The storage and management of geospatial information (both spatial and non-spatial) takes place in the ArcMap environment and pre-defined database files.

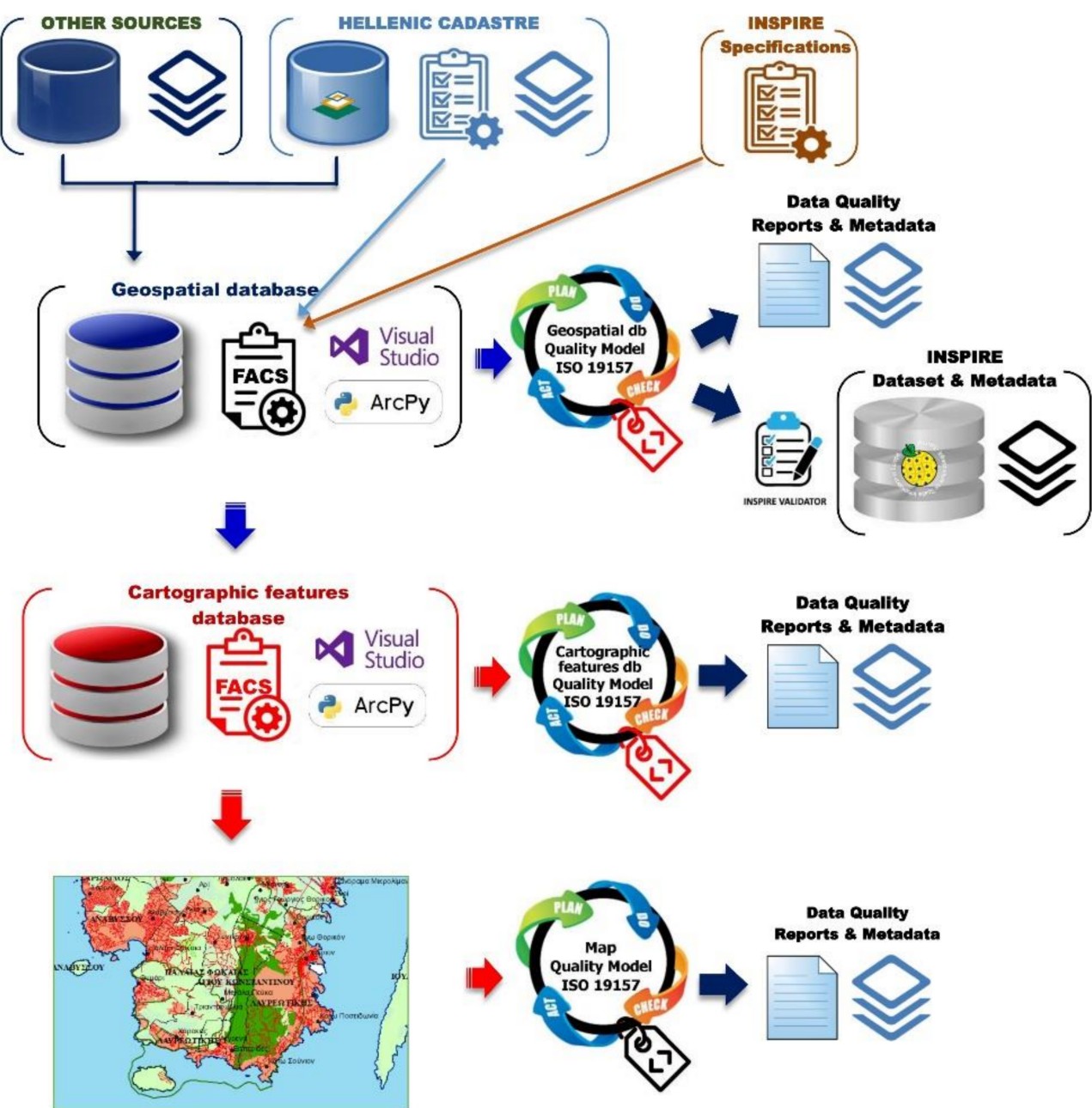

**Figure 2.** 25K map application workflow.

As the area of study and application of the methodology, part of East Attica (Figure 3) was selected. This area was selected due to the fact that it includes almost all kinds of features and covers a 434 km$^2$ area attributed to five 1:25,000-scale map sheets.

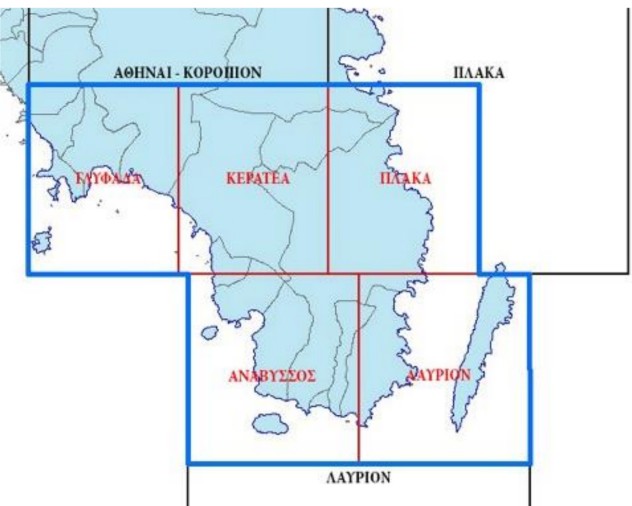

**Figure 3.** Study area.

*5.1. Geospatial Data and Map Specifications*

The methodology for the development of a quality model requires the availability of technical specifications for products generated from geospatial data. The compilation of map specifications was carried out in the following steps:

i.　The specifications of the 1:25,000 maps describing all the entities depicted on the specific map category were compiled from various European cartographic organizations and subsequently analyzed. Selected specifications from ADV of Germany, Instituto Geographico Nacional of Spain, Ordnance Survey of United Kingdom and Swisstopo of Switzerland, were used.

ii.　The entities' structure was compared to each other in terms of: (a) their content, (b) their grouping into thematic units, (c) the hierarchy in their representation and (d) their symbology.

iii.　The main categories and individual entities to be included for portrayal on the final map were selected and the list of entities was created. The technical specifications of the INSPIRE Directive were used to encode the entities and select their properties where available [22–30]. The aim was to produce geospatial data in full compliance with the Directive for the population of the database to be set up.

iv.　On the basis of the selected entities, those required to populate the geospatial database were selected from the geospatial database of the Hellenic Cadastre (HC). As the data available from the HC did not fully cover the requirements, some entities/properties were selected from other sources.

For those entities, where technical specifications were available, compliance with the INSPIRE Directive was chosen for each category and entity [22–30]. For each entity, the use of the common names of entities and their properties, the use of the set of mandatory properties per entity, the inclusion of non-mandatory properties (where deemed necessary) and that of code lists/enumerations was adopted. In addition, complementary properties and corresponding code lists were included in order to make the list more specific to the composite map and to seamlessly migrate the HC data to the geospatial database.

For each category of entities, the corresponding UML class diagram and the list of entities/properties were compiled. Table 4 shows the list of entities/features.

**Table 4.** Feature attributes coding system (FACS).

| Entity/Feature Category | | | |
|---|---|---|---|
| **AdministrativeUnits** | | **Hydrography** | |
| ● | AdministrativeUnit | ●■ | LandWaterBoundary |
| ●■ | AdministrativeBoundary | ● | WatercourseArea |
| **PopulatedPlaces** | | ●■ | WatercourseLine |
| ●■ | PopulatedPlace | ●■ | StandingWater |
| **TransportNetworks** | | ●■ | DamOrWeir |
| ●■ | ERoad | ●■ | Crossing |
| ●■ | Road | ●■ | Falls |
| ●■ | RoadTunnels | ●■ | Spring |
| ●■ | RoadBridges | ●■ | Wetland |
| ● | RoadArea | **LandUse** | |
| ● | RoadNode | ●■ | ExistingLandUseDataSet |
| ●■ | RailwayLine | ● | ExistingLandUseObject |
| ●■ | RailwayTunnels | **ProtectedSites** | |
| ●■ | RailwayBridges | ●■ | ProtectedSite |
| ● | RailwayArea | **Topography** | |
| ● | RailwayNode | ●■ | ContourLine |
| ●■ | RailwayStationNode | ●■ | Mine |
| ●■ | AerodromeArea | **SpotElevation** | |
| ●■ | PortArea | ●■ | SpotElevation |
| **NamedPlaces** | | **CableLinks** | |
| ●■ | NamedPlace | ●■ | CablewayLink |
| **GeneralFeatures** | | ●■ | CablewayNode |
| ●■ | GeneralFeature | | |

Entities marked with ● refer to the content of the geospatial database and features marked with ■ refer to the features of the cartographic database. Topographic symbols related to the terrain relief were not included in the list, as the background of the map was chosen to be displayed as a raster surface with shading of elevation zones (hillshade).

5.1.1. Development and Implementation of Quality Models

Following the building of the FACS for each database, the conceptual, logical and physical model of the data was compiled. Quality information refers to the entity/feature level. The quality model formulates the specification of quality requirements at the entity level, detects the sources of potential errors affecting data quality and identifies the metrics required to quantify quality.

Based on the requirements of the specification and taking into account the requirements of map users, the quality requirements for each individual entity were identified. Initially, the basic quality requirements for each entity as described in the INSPIRE technical specifications were adopted [31]. These were subsequently supplemented and enriched with additional quality requirements. For each quality requirement, quality elements were identified and selected. For each quality element the quality measures and their evaluation methods were identified. Provided that each quality requirement could be checked with more than one quality element, quality measures and evaluation methods, the choice of the most appropriate combination lies with their evaluator. The quality parameters were selected, analyzed and compared with the knowledge of the evaluator and the applicability of these parameters to this dataset was decided. On the basis of the above, the entity/feature-level quality model was developed using ISO 19157:2013 [1,32,33]. The model structure proposed by the European Spatial Data Infrastructure Network program (ESDIN) was adopted [19] (see Tables 5–7).

**Table 5.** Quality model of the geospatial database (part).

| FEATURE TYPE & Attribute | Geospatial Database—QUALITY MODEL ISO 19157 | | | | | | | | | | | | | | |
| --- | --- | --- | --- | --- | --- | --- | --- | --- | --- | --- | --- | --- | --- | --- | --- |
| | QUALITY ELEMENTS | | | | | | | | | | | | | | |
| | COMPLETENESS | | LOGICAL CONSISTENCY | | | | POSITIONAL ACCURACY | | | TEMPORAL ACCURACY | | | THEMATIC ACCURACY | | |
| | COMMISSION | OMISSION | CONCEPTUAL CONSISTENCY | DOMAIN CONSISTENCY | FORMAT CONSISTENCY | TOPOLOGICAL CONSISTENCY | ABSOLUTE ACCURACY | RELATIVE ACCURACY | GRIDDED DATA ACCURACY | ACCURACY OF A TIME MEASUREMENT | TEMPORAL CONSISTENCY | TEMPORAL VALIDITY | CLASSIFICATION CORRECTNESS | NON-QUANTITATIVE ATTRIBUTE CORRECTNESS | QUANTITATIVE ATTRIBUTE ACCURACY |
| AdministrativeUnit | Error count id 2 | Error count id 6 / Error count id 6 | Error count id 9 | | Error indicator id 119 | | | | | | | | Error count id 60 | | |
| inspireId | | | | Error indicator id 14 | | | | | | | | | | | |
| country | | | | Error indicator id 14 | | | | | | | | | | | |
| geometry | Error count id 4 | | Error count id 11 | | | Error count id 25 id 26 id 27 | Id 39 | | | | | | | | |
| name | | | | | | | | | | | | | Error count id 60 | Error count id 65 | |
| nationalCode | | | | Error indicator id 14 | | | | | | | | | | Error count id 65 | |
| HCCode | | | | Error indicator id 14 | | | | | | | | | | Error count id 65 | |
| nationalLevel | | | | Error indicator id 14 | | | | | | | | | | Error count id 65 | |
| nationalLevelName | | | | Error indicator id 14 | | | | | | | | | | Error count id 65 | |
| surfaceArea | | | | | | | | | | | | | | | LE99.8 id 73 |
| beginLifespanVersion | | | | | | | | | | | | | | | |
| endLifespanVersion | | | | | | | | | | | | | | | |
| AdministrativeBoundary | Error count id 2 | Error count id 6 | Error count id 9 | | Error indicator id 119 | | | | | | | | | | |
| inspireId | | | | | | | | | | | | | Error count id 65 | | |
| country | | | | | | | | | | | | | Error count id 65 | | |
| geometry | Error count id 4 | | | | | Error count id 21 id 23 id 24 id 27 | | | | | | | | | |
| nationalLevel | | | | | | | | | | | | | Error count id 65 | | |
| length | | | | | | | | | | | | | | | LE99.8 id 73 |
| beginLifespanVersion | | | | | | | | | | | | | | | |
| endLifespanVersion | | | | | | | | | | | | | | | |

**Table 6.** Quality model of cartographic database (part).

| FEATURE TYPE & Attribute | COMPLETENESS | | LOGICAL CONSISTENCY | | | | POSITIONAL ACCURACY | | | TEMPORAL ACCURACY | | | THEMATIC ACCURACY | | |
|---|---|---|---|---|---|---|---|---|---|---|---|---|---|---|---|
| | COMMISSION | OMISSION | CONCEPTUAL CONSISTENCY | DOMAIN CONSISTENCY | FORMAT CONSISTENCY | TOPOLOGICAL CONSISTENCY | ABSOLUTE ACCURACY | RELATIVE ACCURACY | GRIDDED DATA ACCURACY | ACCURACY OF A TIME MEASUREMENT | TEMPORAL CONSISTENCY | TEMPORAL VALIDITY | CLASSIFICATION CORRECTNESS | NON-QUANTITATIVE ATTRIBUTE CORRECTNESS | QUANTITATIVE ATTRIBUTE ACCURACY |
| AdministrativeBoundary | Error count id 2 | Error count id 6 | Error count id 9 | | Error indicator id 119 | | | | | | | | | | |
| inspireId | | | | | | | | | | | | | Error count id 65 | | |
| country | | | | | | | | | | | | | Error count id 65 | | |
| geometry | Error count id 4 | | | | | Error count id 21 id 23 id 24 id 27 | Error count id 30 | | | | | | | | |
| nationalLevel | | | | | | | | | | | | | Error count id 65 | | |
| length | | | | | | | | | | | | | | | LE99.8 id 73 |
| beginLifespanVersion | | | | | | | | | | | | | | | |
| endLifespanVersion | | | | | | | | | | | | | | | |

**Table 7.** Quality model of map (part).

| FEATURE TYPE & Attribute | COMPLETENESS | | LOGICAL CONSISTENCY | | | | POSITIONAL ACCURACY | | | TEMPORAL ACCURACY | | | THEMATIC ACCURACY | | |
|---|---|---|---|---|---|---|---|---|---|---|---|---|---|---|---|
| | COMMISSION | OMISSION | CONCEPTUAL CONSISTENCY | DOMAIN CONSISTENCY | FORMAT CONSISTENCY | TOPOLOGICAL CONSISTENCY | ABSOLUTE ACCURACY | RELATIVE ACCURACY | GRIDDED DATA ACCURACY | ACCURACY OF A TIME MEASUREMENT | TEMPORAL CONSISTENCY | TEMPORAL VALIDITY | CLASSIFICATION CORRECTNESS | NON-QUANTITATIVE ATTRIBUTE CORRECTNESS | QUANTITATIVE ATTRIBUTE ACCURACY |
| AdministrativeBoundary | Error count id 2 | Error count id 6 | | | | | | | | | | | Error count id 6 | | |
| hierarchyLevel | | | | | | | | | | | | | | Error count id 65 | |
| lineSymbol | | | | | | | | | | | | | | Error count id 65 | |
| lineWidth | | | | | | | | | | | | | | Error count id 65 | |
| lineColor | | | | | | | | | | | | | | Error count id 65 | |

For quality checks included in the quality model, fully automated (full inspection) or semi-automated (sampling inspection) were implemented through the software application developed. In the Tables 5–7, the colored cells describe the quality element as applied at the entity/feature or the entity/feature attribute level depending on the respective QM. The number in the cell indicates the standardized data quality measure code chosen to quantify the quality of the quality element as contained in the Annex D of ISO 19157:2013 [1]. To quantify the quality measure, there were three procedures provided for the inspections:

✓ Attribute inspection by sampling according to ISO 2859-1: 1999 [34] (yellow cells);
✓ Variable inspection by sampling according to ISO 3951-1:2013 [35] for the sample size and FGDC standard [36,37] for the distribution of the check points (green cells);
✓ Full inspection (orange cells).

Table 5 shows a part of the quality model of the geospatial database. It includes the quality elements and measures referring to: (a) the dataset of administrative units (AU) that were inserted as polygons into the geodatabase from the reference data of the Hellenic Cadastre, and (b) the dataset of administrative boundaries (AB) as they resulted from the processing of the administrative units dataset (automated transition from AU to AB within the software application).

Table 6 lists part of the cartographic database quality model. It includes the quality elements and measures related to the dataset of administrative boundaries as derived after importing and generalizing the administrative boundaries dataset of the geospatial geodatabase.

Table 7 shows the part of the quality model for a section of the map. It includes the quality elements and measures related to the dataset of administrative boundaries as they will be portrayed on the map.

From Tables 5 and 6, it is derived that the quality parameters and quality measures chosen in the compilation of the quality models of the geospatial database and the cartographic database are similar. The similar quality models are the result of a methodological choice to include from the outset in the geospatial database also entities/properties of the cartographic database that are essential for the composition of the final map. For example, in the entity related to dams, properties related to the representation of the symbol on the map are added, such as DamFaceDirection and DamFaceAngle.

This choice enabled the evaluator to perform all the checks of the cartographic database in a fully automated way, using as reference data the entities/properties of the geospatial database. After comparing the quality results, any differences that arose were due to the transition and generalization of entities from the geospatial database to the cartographic features database and were related to residual errors.

### 5.1.2. Creation of the Geospatial Database

For the most part, the geospatial entities and the descriptive information to be used to "populate" the geospatial database came from the Digital Cadastral Data Base (DCDB) of the Hellenic Cadastre. The Hellenic Cadastre stores geospatial data at a scale of 1:1000 for urban areas and 1:5000 for rural and other areas. The spatial entities of the DCDB are structured in subcategories, kept at the level of municipalities and are available in a shapefile format. The descriptive information is fully linked to the spatial entities and is available in MS Access format files. The creation of the geodatabase and its population with entities and attribute values was mostly performed automatically (with python scripts) using the software application developed (see step 4 below).

The information originating from other sources than the cadastre mainly concerned attribute values and was encoded in digital reference files (shapefile and MS Access accdb).

The methodology for populating the database using the software application developed follows seven steps (Figure 4):

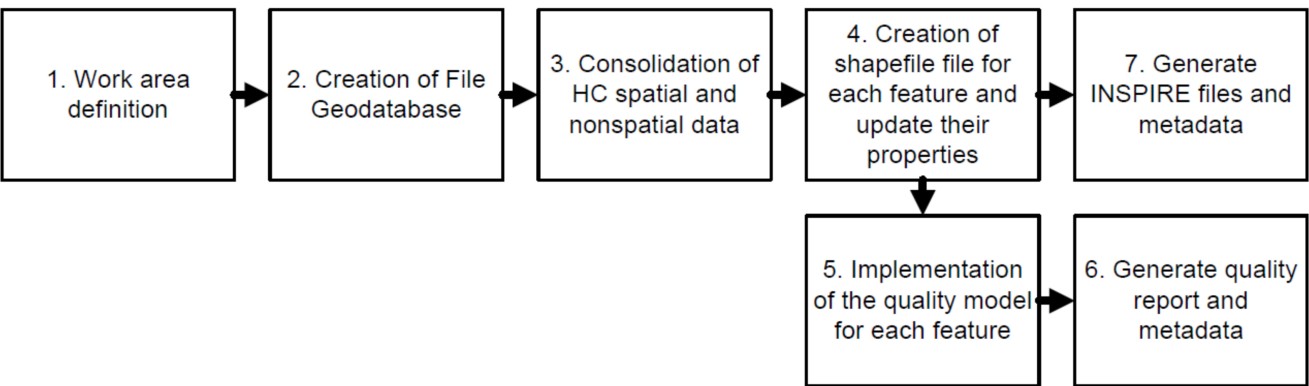

**Figure 4.** Workflow of geospatial database creation.

1st step: The user enters into the application a closed polygon that delineates the area to be covered by the map. Then, the application automatically generates a list of codes of the municipalities located within the delineated polygon.

2nd step: The geospatial database is automatically created at the spatial entity level per category, based on FACS and the input and reference geospatial data are stored automatically.

3rd step: Automated consolidation of HC and other sources spatial and non-spatial data. Single files per entity are automatically produced for the whole work area.

4th step: A shapefile is automatically created for each entity and updated with the corresponding spatial entities and their attributes as defined in FACS. For each entity, a script was developed in the Python programming language (using the ArcPy site package), which was used for the automated transition of the entities to the geospatial database and for updating their attributes.

The following is an example of updating one of the "RoadNode" entity attributes that define the hierarchy of road centerline representations at multi-level nodes. It is implemented in three stages:

i.      Automated node detection from the road line network (Figure 5).

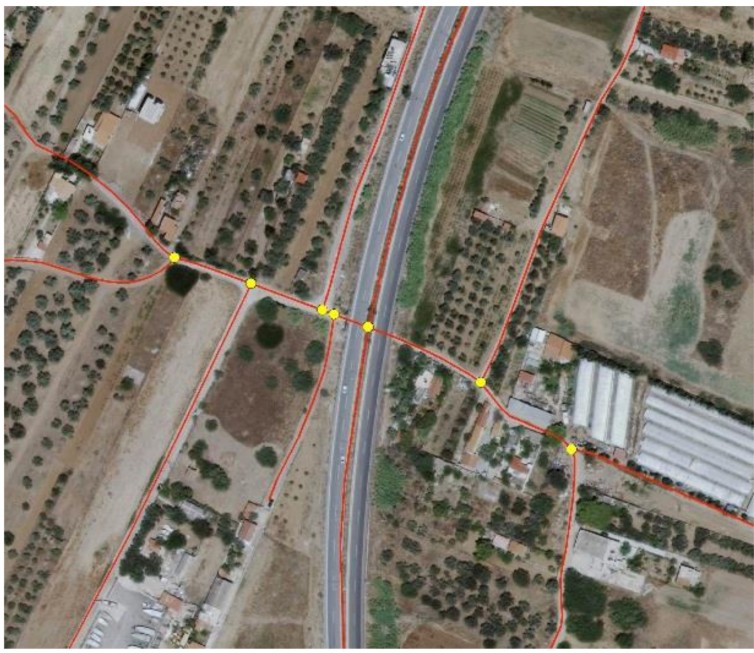

**Figure 5.** Automated node detection.

ii.    Automated node detection with ≥4 line segments involved, marking areas for processing and cutting sections of road centerlines for processing (Figure 6).

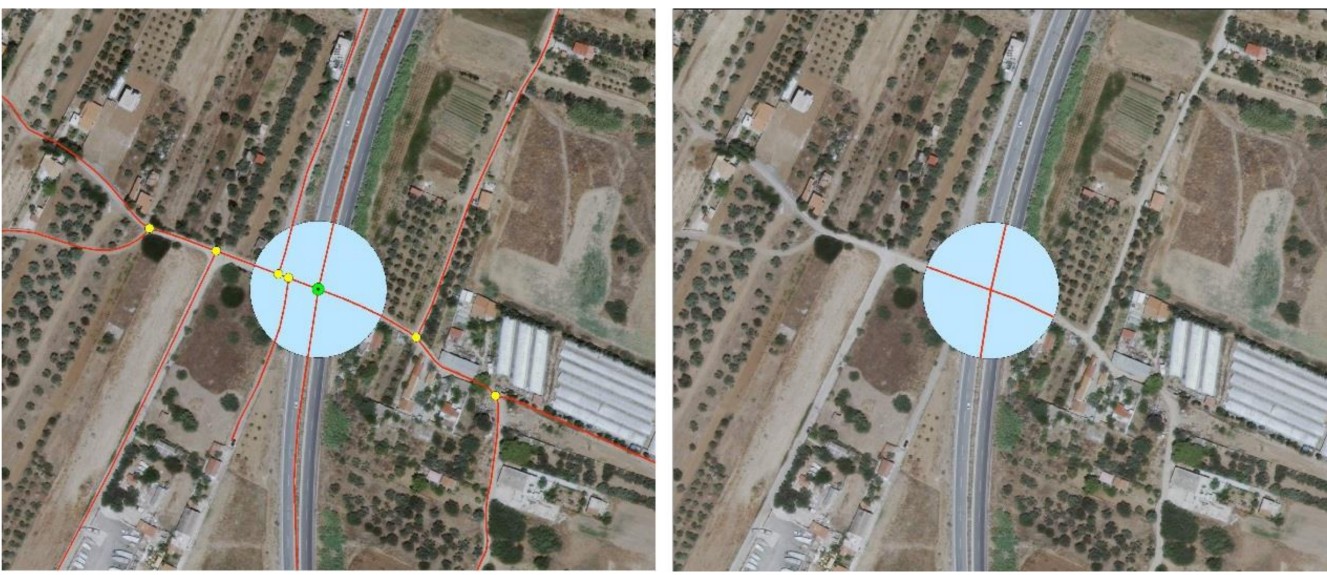

**Figure 6.** Selection of road sections for editing.

iii.    Automated slope map extraction using the digital elevation model and checking of the maximum slope attribute value (Figure 7). The maximum value of the attribute indicates whether the node is multi-level or not and determines which road centerline passes over.

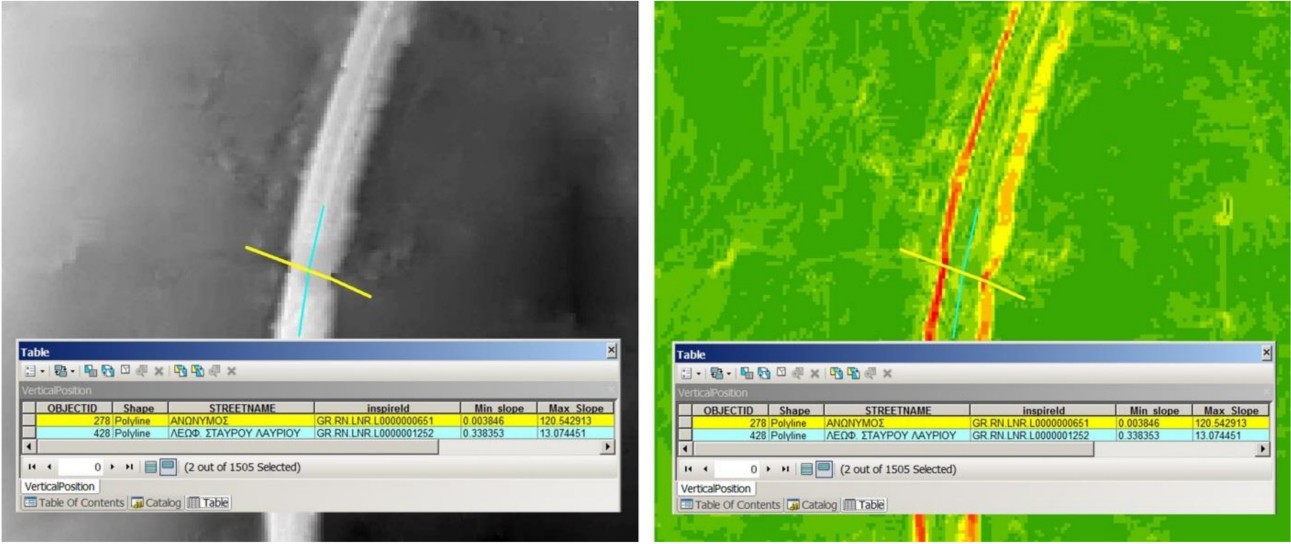

**Figure 7.** Check max slope attribute value from slope map.

5th step: Implementation of the quality model. Automated and semi-automated quality inspection of spatial and descriptive information and acquisition of quantified quality information for each quality parameter of the QM. If one of the populated entities has appropriate and documented quality information, it is not necessary to inspect it using the software application. In this case, the evaluator will manually enter the existing quality measures in a specific table within the application. As the geodatabase update is implemented by processing and transforming the input data, it is recommended to perform all the automated checks foreseen in the quality model.

6th step: Automated generation of quality reports and metadata related to the results of quality inspection at entity level. The information structure is stored in full compliance with ISO standards 19157:2013 [1], ISO 19115-1:2014 [38] and ISO 19915-3:2016 [39].

7th step: Automated generation of the files of spatial entities in accordance with the technical specifications of the INSPIRE Directive and the corresponding metadata. At this stage, the spatial entity files are stripped off the attributes added to accommodate the data model specifications that are not included in the technical specifications of the INSPIRE Directive.

### 5.1.3. Creation of the Cartographic Features Database

The entities resulting from the previous stage of the work are transferred into the cartographic database that will be used for the production of a topographic map at the 1:25,000 scale. According to standard practice, an intermediate "transition" database will have to be created that will contain only the entities and their attributes required by the map's FACS. Since the whole process is executed automatically, the transition of the geospatial database entities, the deletion of non-useful attributes and their generalization were implemented in a single step of the process (step 2). This methodological approach was chosen because the full automation of the process has a minimal impact on the implementation time.

The methodology for the transition of entities to the cartographic features database using the software application developed follows four steps (Figure 8):

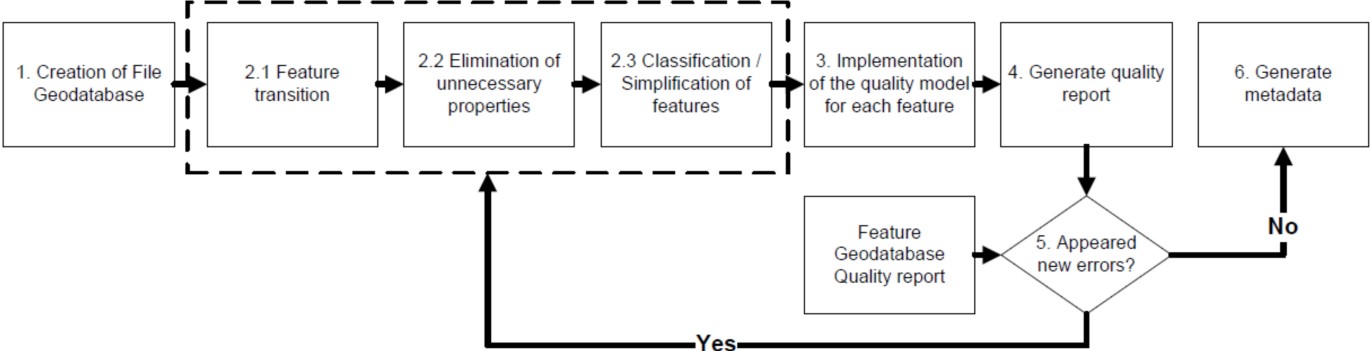

**Figure 8.** Workflow of cartographic features database creation.

1st step: The cartographic database is automatically created at the spatial entity level per category, based on FACS.

2nd step: Transition of geospatial database entities to the created cartographic database. It is implemented in three individual steps using a single python script in the following order:

i.    Transition of entities: A shapefile file is generated automatically for each entity and updated with their respective spatial entities and their attributes.

ii.    Elimination of non-useful entity attributes: Attributes that are not useful in map production are removed. Entities, due to the need of complying with the INSPIRE Directive, include many attributes that are not necessary for the composition of the map.

iii.    Generalization: Apply generalization rules to entities where necessary based on the scale of the map produced. This includes the classification and simplification of linear and areal entities.

An example of simplification of the 'LandWaterBoundary' entity is given below for the parts representing the natural coastline. The generalization methodology is based on [40]. It is implemented in three stages:

i.    Automated line simplification using bend simplification algorithm with a 15 m tolerance (Figure 9). Automated evaluation of simplification using 12.5 m buffer zone on both sides (Figure 10).

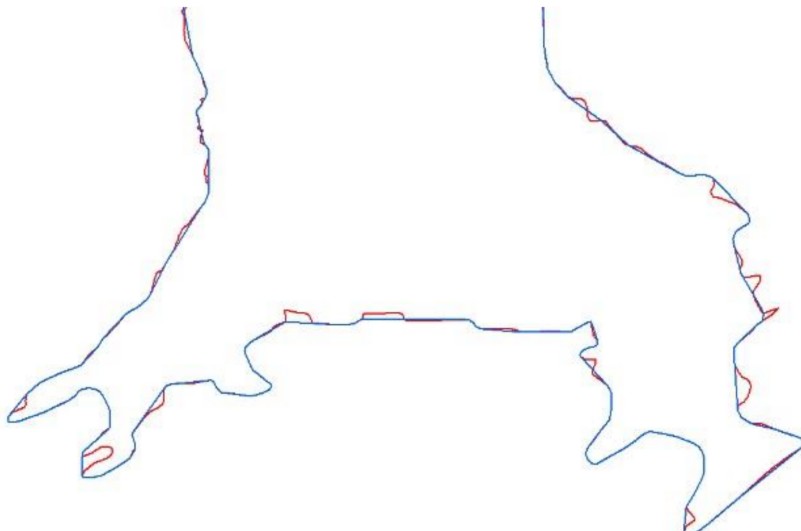

**Figure 9.** Bend simplification.

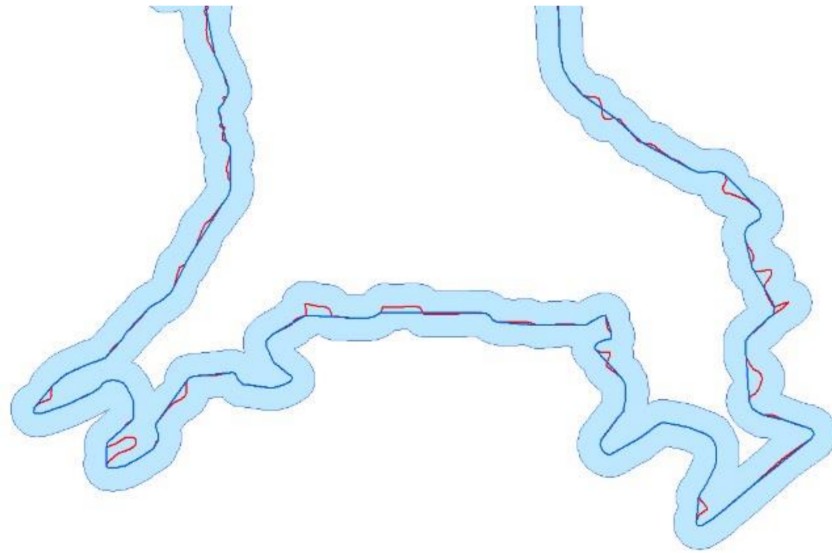

**Figure 10.** Evaluation of bend simplification.

ii. Automated line simplification using the point removal method (Figure 11). As a basic rule, the small sections of the line are integrated into new ones with a length between 7.5–10 m.

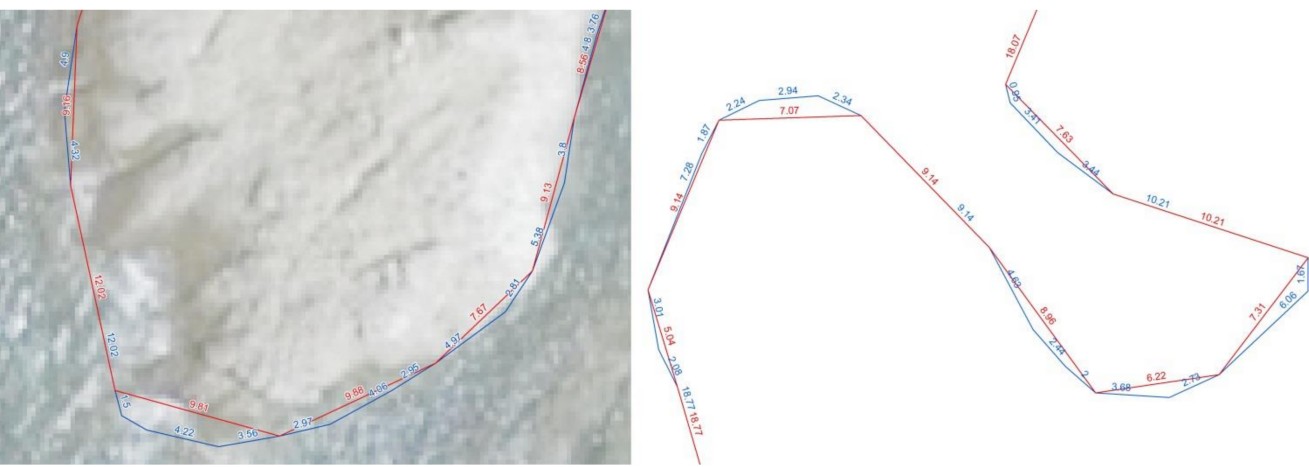

**Figure 11.** Simplification using point removal.

iii.    Island management

● Automated simplification with conversion of islands with area <625 m$^2$ into point entities (Figures 12 and 13). The transformed islands are shown as red dots in Figure 13.

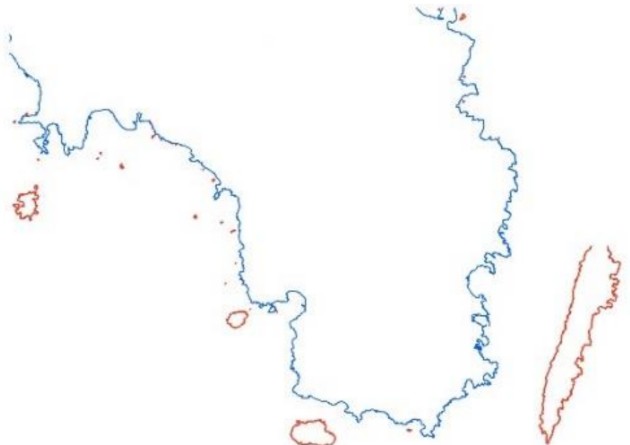

**Figure 12.** Island detection.

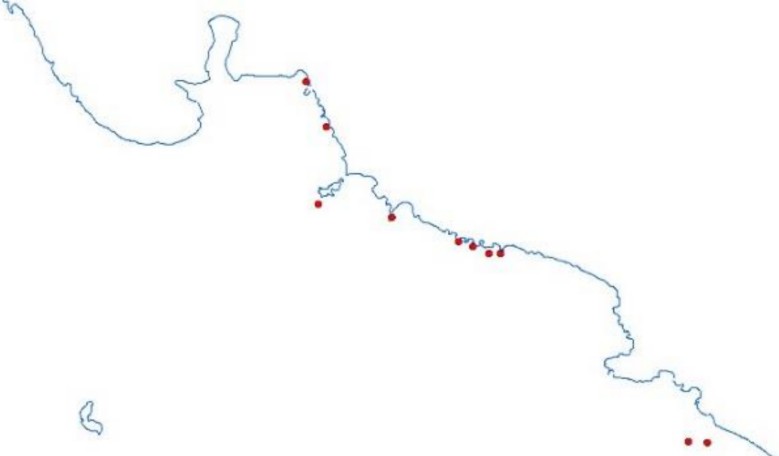

**Figure 13.** Conversion to point entities.

- Aggregation of islands located in proximity to the mainland coastline with respect to the chart compilation scale. Integration of the islands that are located within a distance of less than 25 m from the coastline (Figure 14).
  - (a) Calculation of the distance of islands from the closest coastline.
  - (b) Selection of islands, located within a distance less than 25 m from the coastline and demarcation of minimum bounding area.
  - (c) The islands/islets amalgamate to the closest coastline.

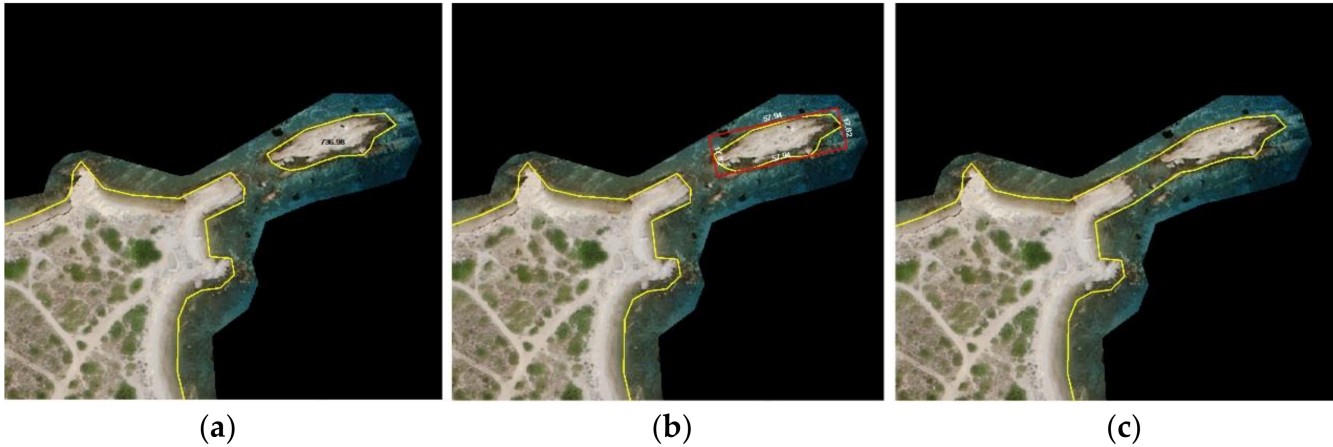

(**a**)        (**b**)        (**c**)

**Figure 14.** Integration of islands with coastline.

As a result of the software application on the above, the cartographic database includes the features in the way they will be portrayed on the final map.

3rd step: Implementation of the quality model. Automated and semi-automatic quality inspection of spatial and descriptive information and acquisition of quantified quality information for each quality parameter of QM.

4th step: Automated generation of quality reports related to the results of quality inspection at the entity level. The information structure is stored in full compliance with the ISO 19157:2013 [1] standard.

5th step: Benchmarking of quality results at the entity and quality element level. If the evaluation results contain new errors, these were obviously created during the transition or/and generalization of the data. In this case, the producer can review the methods used (step 2) so that new errors are eliminated or minimized.

6th step: Automated generation of metadata related to the results of quality inspection at the entity level. The information structure is stored in full compliance with ISO 19115-1:2014 [1], ISO 19115-1:2014 [38] and ISO 19915-3:2016 [39].

### 5.1.4. Map Composition

The cartographic composition is to consolidate and assign, in accordance with the rules of cartographic symbolization and cartographic design, the set of the thematic levels of the cartographic data collected and processed in the previous phases of the process [41]. In addition, to complete the map composition, all the elements necessary for the map layout should be added, such as nomenclature, grid, box, graphic scale, legend, title, etc. The principles of cartographic design refer to the characteristics that a map should have as a product of graphic design [42]. These characteristics are: (a) symbol legibility, (b) visual contrast, (c) image-background organization and (d) hierarchical organization.

The map depicts the content of the cartographic database and the spatial relationships between the entities. Map composition briefly includes the following tasks (Figure 15):

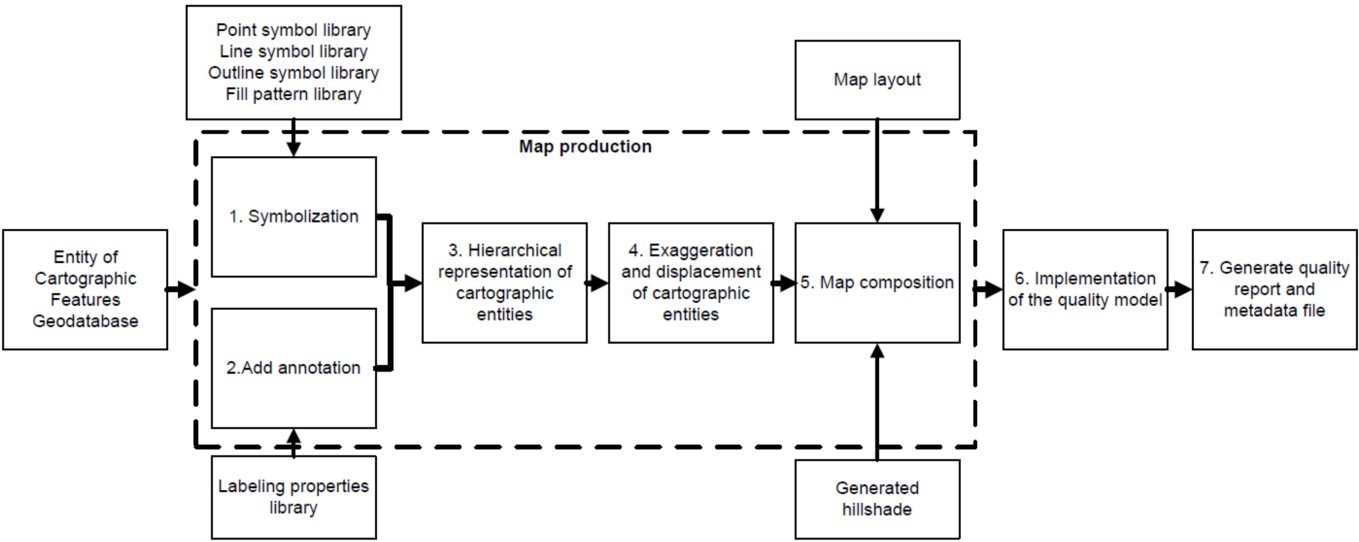

**Figure 15.** Workflow of map composition.

1st step: Symbolization. Each element is automatically assigned the predefined symbol from the corresponding symbol library created for the 1:25,000 map scale. It concerns the representation of points (point symbol library, point symbol, size, rotation, color), lines (line symbol library, line symbol type, width, color) and areas (outline symbol library, outline symbol type, outline width, outline color, fill pattern library, fill color, fill pattern).

2nd step: Automated addition of geographical names. Representation of labels using the properties of the corresponding library, predefined for each type of geographical name. The properties define the font's name, width factor, height, style, slope, alignment, rotation and color.

3rd step: Automated hierarchical representation of cartographic features, according to the value of the corresponding property assigned to the hierarchy level.

4th step: Semi-automated exaggeration and displacement of cartographic features to make the map visually balanced.

5th step: Synthesis of the final map. Automated addition of shaded terrain as a background for the rendering of cartographic data and cutting the area into sections based on a predefined map sheet distribution. Automated application of the predefined map layout created based on a predefined template.

6th step: Implementation of the quality model. Visual inspection of cartographic features and acquisition of quantified quality information for each quality parameter of the QM.

7th step: Automated generation of quality reports related to the results of quality inspection at the cartographic feature level. The information structure is stored in full compliance with the ISO 19157:2013 [1] standard.

## 6. Results

The application developed includes a specific functionality to assist in the evaluation of each distinct set of entities/cartographic features. The results of the evaluation are stored at the entity/feature level in the database in a special MS Access table. The evaluator can then automatically export the evaluation results, in the form of a quality report and/or in the form of metadata (XML format files), based on the requirements of the ISO 19157:2013 standard [1]. Where the data have appropriate and documented quality information at the entity level, the evaluator is not required to perform the inspection. In this case, the functionality of the software application gives the evaluator the option to record the quality measurements manually. In the authors' opinion, it is strongly recommended that in these cases all automated checks be performed, as the cost to the execution time is limited.

After the inspection checks are completed, the software application enables the evaluator to automatically export the quality results at the entity level in the form of a quality report (Figure 16) and/or metadata.

**Figure 16.** Quality report (part).

The cadastral data used have documented but unpublished quality. The only published quality documentation relates to the positional accuracy of the geospatial information [43]. Although the quality information could be obtained from the HC quality results, for the requirements of the research and the evaluation of the functionality of the software application, the inspections foreseen in the quality model were carried out. A comparative check between similar quality results was then carried out in order to check the correctness of the application.

An example of the application of the quality models is given below. The model was selected for the administrative units dataset at all stages of the map composition.

### 6.1. Geospatial Database

Tables 8 and 9 shows the results of the implementation of the quality model as extracted from the software application.

**Table 8.** Implementation results of the QM of the geospatial database—administrative unit.

| ID | FeatureType/ Attribute | DQ ELEMENT | DQ Sub_ELEMENT | NameOf-Measure | Measure-Identification | DQ_Quantitative-Result | Result-ValueType |
|----|----------------------|------------|----------------|----------------|------------------------|------------------------|------------------|
| 1 | AdministrativeUnit | Completeness | Commission | Error count | 2 | 0 | Integer |
| 2 | AdministrativeUnit | Completeness | Omission | Error count | 6 | 0 | Integer |
| 3 | AdministrativeUnit | Logical consistency | Conceptual Consistency | Error count | 9 | 0 | Integer |

**Table 8.** *Cont.*

| ID | FeatureType/Attribute | DQ ELEMENT | DQ Sub_ELEMENT | NameOf-Measure | Measure-Identification | DQ_Quantitative-Result | Result-ValueType |
|---|---|---|---|---|---|---|---|
| 4 | AdministrativeUnit | Logical consistency | Format consistency | Error indicator | 119 | True | Boolean |
| 5 | AdministrativeUnit | Thematic accuracy | Classification Correctness | Error count | 60 | 0 | Integer |
| 6 | inspireId | Logical consistency | Domain consistency | Error indicator | 14 | True | Boolean |
| 7 | country | Logical consistency | Domain consistency | Error indicator | 14 | True | Boolean |
| 8 | geometry | Completeness | Commission | Error count | 4 | 0 | Integer |
| 9 | geometry | Logical consistency | Conceptual Consistency | Error count | 11 | 0 | Integer |
| 10 | geometry | Logical consistency | Topological Consistency | Error count | 25 | 3 | Integer |
| 11 | geometry | Logical consistency | Topological Consistency | Error count | 26 | 0 | Integer |
| 12 | geometry | Logical consistency | Topological Consistency | Error count | 27 | 0 | Integer |
| 13 | geometry | Positional accuracy | Absolute accuracy | Root mean square error | 39 | 1.22 | Meters |
| 14 | name | Thematic accuracy | Non-quantitative attribute correctness | Error count | 60 | 0 | Integer |
| 15 | name | Thematic accuracy | Non-quantitative attribute correctness | Error count | 65 | 0 | Integer |
| 16 | nationalCode | Thematic accuracy | Non-quantitative attribute correctness | Error count | 65 | 0 | Integer |
| 17 | nationalCode | Logical consistency | Domain consistency | Error indicator | 14 | True | Boolean |
| 18 | HCCode | Thematic accuracy | Non-quantitative attribute correctness | Error count | 65 | 0 | Integer |
| 19 | HCCode | Logical consistency | Domain consistency | Error indicator | 14 | True | Boolean |
| 20 | nationalLevel | Thematic accuracy | Non-quantitative Attribute correctness | Error count | 65 | 0 | Integer |
| 21 | nationalLevel | Logical consistency | Domain consistency | Error indicator | 14 | True | Boolean |
| 22 | nationalLevelName | Thematic accuracy | Non-quantitative Attribute correctness | Error count | 65 | 0 | Integer |
| 23 | nationalLevelName | Logical consistency | Domain consistency | Error indicator | 14 | True | Boolean |
| 24 | surfaceArea | Thematic accuracy | Quantitative attribute correctness | LE99.8 | 73 | True | Boolean |

**Table 9.** Implementation results of the QM of the geospatial database—administrative boundary.

| ID | FeatureType/Attribute | DQ ELEMENT | DQ Sub_ELEMENT | NameOf-Measure | Measure-Identification | DQ_Quantitative-Result | Result-ValueType |
|---|---|---|---|---|---|---|---|
| 1 | AdministrativeBoundary | Completeness | Commission | Error count | 2 | 0 | Integer |
| 2 | AdministrativeBoundary | Completeness | Omission | Error count | 6 | 0 | Integer |
| 3 | AdministrativeBoundary | Logical consistency | Conceptual Consistency | Error count | 9 | 0 | Integer |
| 4 | AdministrativeBoundary | Logical consistency | Format Consistency | Error indicator | 119 | True | Boolean |
| 5 | inspireId | Thematic accuracy | Non-Quantitative Attribute Correctness | Error count | 65 | 0 | Integer |
| 6 | country | Thematic accuracy | Non-quantitative Attribute correctness | Error count | 65 | 0 | Integer |
| 7 | geometry | Completeness | Commission | Error count | 4 | 0 | Integer |
| 8 | geometry | Logical consistency | Topological Consistency | Error count | 21 | 0 | Integer |
| 9 | geometry | Logical consistency | Topological Consistency | Error count | 23 | 0 | Integer |
| 10 | geometry | Logical consistency | Topological Consistency | Error count | 24 | 0 | Integer |
| 11 | geometry | Logical consistency | Topological Consistency | Error count | 27 | 0 | Integer |
| 12 | nationalLevelName | Thematic accuracy | Non-quantitative Attribute correctness | Error count | 65 | 0 | Integer |
| 13 | length | Thematic accuracy | Quantitative attribute Correctness | LE99.8 | 73 | True | Boolean |

The administrative division of Greece and digital orthophoto maps were used as reference data for the sample inspections. Commenting on the quality results, it is shown that in terms of completeness and thematic accuracy, as expected, the results are zero. By contrast, when checking for logical consistency, three cases of invalid slivers were detected. The data were examined and it was found that three slivers were intrinsic to the data and did not arise due to their transition to the geospatial database (Table 8).

The administrative boundaries of the municipalities resulted from the transition of the dataset administrative unit entities. With an already inspected reference dataset, we can adopt compliance levels on specific quality measures (Table 9). If any of the generated data do not comply with the default compliance level, the evaluator may revise the transition methodology of that entity and re-run the inspection. As the entities of the Administrative Unit dataset were used as reference data for the administrative boundary checks, the checks were fully automated.

### 6.2. Cartographic Features Database

The geospatial database entities were used as reference data, and for this reason the checks were fully automated. Again, as the import information was of known quality, the producer could pre-define compliance levels. The quality results show that the slivers identified in the administrative unit dataset were eliminated (Table 10). This was achieved because their existence and their exact location was known from the beginning, so the evaluator could choose an appropriate methodology in the transition of entities in order to eliminate them.

**Table 10.** Implementation results of the QM of the cartographic features geodatabase—administrative boundary.

| ID | FeatureType_Attribute | DQ ELEMENT | DQ Sub_ELEMENT | NameOf-Measure | Measure-Identification | DQ_Quantitative-Result | Result-ValueType |
|---|---|---|---|---|---|---|---|
| 1 | AdministrativeBoundary | Completeness | Commission | Error count | 2 | True | Integer |
| 2 | AdministrativeBoundary | Completeness | Omission | Error count | 6 | True | Integer |
| 3 | AdministrativeBoundary | Logical consistency | Conceptual Consistency | Error count | 9 | 0 | Integer |
| 4 | AdministrativeBoundary | Logical consistency | Format consistency | Error indicator | 119 | True | Boolean |
| 5 | inspireId | Thematic accuracy | Non-quantitative Attribute Correctness | Error count | 65 | 0 | Integer |
| 6 | country | Thematic accuracy | Non-quantitative attribute correctness | Error count | 65 | 0 | Integer |
| 7 | geometry | Completeness | Commission | Error count | 4 | 0 | Integer |
| 8 | geometry | Logical consistency | Topological Consistency | Error count | 21 | 0 | Integer |
| 9 | geometry | Logical Consistency | Topological Consistency | Error count | 23 | 0 | Integer |
| 10 | geometry | Logical consistency | Topological Consistency | Error count | 24 | 0 | Integer |
| 11 | geometry | Logical consistency | Topological Consistency | Error count | 27 | 0 | Integer |
| 12 | geometry | Positional accuracy | Absolute accuracy | Number of positional uncertainties above a given threshold | 30 | 9.28 | Meters |
| 13 | nationalLevel | Thematic accuracy | Non-quantitative attribute correctness | Error count | 65 | 0 | Integer |
| 14 | length | Thematic accuracy | Quantitative attribute correctness | LE99.8 | 73 | True | Boolean |

### 6.3. Map

The sampling inspection is carried out visually on the final map. At the moment it has not yet been executed and therefore in Table 11 the values of the quality result (column "DQ_QuantitativeResult") are not filled in. The reference data for the inspection are: (a) cartographic base for completeness, thematic accuracy, hierarchical layers, etc., (b) orthophoto maps for classification correctness, multilevel nodes, unpaved roads, etc., (c) symbol library for symbol representation and (d) labeling properties library for geographical names.

**Table 11.** Implementation results of the QM of the map—administrative boundary.

| ID | FeatureType_Attribute | DQ ELEMENT | DQ Sub_ELEMENT | NameOf-Measure | Measure-Identification | DQ_Quantitative-Result | Result-ValueType |
|---|---|---|---|---|---|---|---|
| 1 | AdministrativeBoundary | Completeness | Commission | Error count | 2 | | Integer |
| 2 | AdministrativeBoundary | Completeness | Omission | Error count | 6 | | Integer |
| 3 | AdministrativeBoundary | Thematic accuracy | Classification Correctness | Error count | 60 | | Integer |
| 4 | hierarchyLevel | Thematic accuracy | Non-quantitative attribute correctness | Error count | 65 | | Integer |
| 5 | lineSymbol | Thematic accuracy | Non-quantitative attribute correctness | Error count | 65 | | Integer |
| 6 | lineWidth | Thematic accuracy | Non-quantitative attribute correctness | Error count | 65 | | Integer |
| 7 | lineColor | Thematic accuracy | Non-quantitative attribute correctness | Error count | 65 | | Integer |

## 7. Discussion

This paper proposes the adoption of quality models as fundamental tools of an integrated environment for monitoring and documenting quality at all stages of the map production line. It also provides guidelines for the creation of a quality model to quantify, monitor and document the quality of geospatial data. As part of the research, an integrated software application for the utilization of the country's cadastral information for the composition of a 1:25,000-scale map was also developed. As the cadastral information is collected and maintained digitally using the tools available with the GIS software, the production of the map was largely automated.

The results of the implementation of the quality model at each individual stage of map production show the significant advantages it offers in quality management.

For the creation of the geospatial database, the workflow includes the acquisition of the required geospatial information, its input and storage in the geodatabase, the manipulation and its processing. In the process of establishing the geospatial database, because of the above operations, errors can occur that affect its quality [44]. These errors, whether random or systematic, will be carried forward to the next stage of the map compilation process, and it is important that they be identified, recognized, quantified and recorded. The result of the application of a quality model to the data of the geospatial database is the acquisition of quantified quality information on specific quality measures and its documentation. The application of the quality model enables the producer to know exactly which entities include errors, as well as their type, their number and their location. An accurate knowledge and mapping of errors enables the cartographer to assess how they will affect its final product and to use an appropriate methodology during the next stage of the process in order to either eliminate them or minimize their impact on the quality of the composite map. In addition, the producer may also set in the quality model compliance criteria at the entity/quality level. By using them, after evaluating the input data, the known quantified quality information enables him/her to decide whether and to what extent the data are suitable for the composite map. If he/she decides to reject them, he/she can choose a new dataset and repeat the evaluation using the quality model.

The cartographic database is derived from the transition of the geospatial data of the geospatial database. The process of transition involves the input and storage of data in the cartographic geodatabase and the simplification/generalization of the geospatial information of the database. Regarding the monitoring of the quality of the cartographic database, in addition to any errors transferred over from the geospatial database, the process of the transition and generalization of information may also create new errors. The best way to evaluate, document and present the quality of the cartographic features database is to adopt a quality model. The quality model of the cartographic database, although based on a different FACS, can nevertheless use in most entities similar quality measures to the geospatial database quality model. The adoption of similar-quality enables the producer to compare them, so that he knows whether a particular error was transferred from the

geospatial database or arose during data transition and generalization. Where the comparison identifies new errors, it shall be given the opportunity to revise the methodology applied with a view to eliminating or minimizing them.

At the cartographic composition stage, generalized features are presented graphically. The process of the transposition of the features of the cartographic database includes their hierarchical rendering as well as the portrayal of lines, points, surfaces, symbols and labels—a task that is likely to generate errors in their portrayal. The adoption of a quality model for the evaluation and documentation of the quality of the produced map enables the producer to fully control the production process and its quality as well as the users with evidence of control and inspection, while providing more information about the final product through metadata.

### 8. Conclusions and Future Research

The outcome of the research confirms that the design and implementation of a quality model at each stage of the process provides a structured framework for defining, evaluating and documenting quality and managing quality information. It also provides an integrated structured quality assurance environment at every stage of the map composition process. It enables the producer to have a high level of control over the production process, to identify and manage errors, and to improve the production process and the quality of a product. At the same time, the use of international standards in the development and implementation of the quality model and the harmonization of quality information with them (a) ensures the interoperability of quality information, and (b) provides an environment for applying consistent and objective quality inspection methods.

Future research may focus on the following:

- To analyze the needs, in terms of quality, of the different users of spatial data, by identifying and effectively recording how many and which of the quality parameters users want to be recorded and how.
- To improve the way in which the results of the quality inspection are recorded and presented so that they are more comprehensible to the average user. ISO 19157:2013 [1], although estimated to be complete and detailed in the documentation of the quality results and their recording, is mainly intended for specialized users.

For different types of data, identification of quality requirements, selection of quality parameters that may be applied for the particular type of data and their definition are required for each quality parameter to be assessed.

**Author Contributions:** Conceptualization, I.K. and L.T.; methodology, I.K. and L.T.; software, I.K.; validation, I.K. and L.T.; formal analysis, I.K.; investigation, I.K.; resources, I.K.; data curation, I.K.; writing—original draft preparation, I.K.; writing—review and editing, L.T.; supervision, L.T. All authors have read and agreed to the published version of the manuscript.

**Funding:** This research received no external funding.

**Institutional Review Board Statement:** Not applicable.

**Informed Consent Statement:** Not applicable.

**Data Availability Statement:** https://maps.gov.gr; https://data.ktimatogio.gr (accessed on 3 May 2022).

**Conflicts of Interest:** The authors declare no conflict of interest.

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
