# Peer review of "An Integrated Environment for Monitoring and Documenting Quality in Map Composition Utilizing Cadastral Data"

_ijgi, doi:10.3390/ijgi11060348_

Round 1

Reviewer 1 Report

This paper introduced their research results. But the relative methods are not clear. The authors said "Automated and semi-automatic quality inspection of spatial and descriptive information and acquisition of quantified quality information for each quality parameter of QM.", but how to quantity the quality information could not be found in this paper. And the design and implementation of a “quality model” based on the ISO 19157:2013 standard is not clear. This paper likes a research report, then  it should be difficult for paper readers to get new knowledge from this paper. The software developed by authors for this research should be uploaded.

Author Response

Dear reviewer,

Thank you for your constructive comments. Please find our response in the following paragraphs: 

“This paper introduced their research results. But the relative methods are not clear. The authors said "Automated and semi-automatic quality inspection of spatial and descriptive information and acquisition of quantified quality information for each quality parameter of QM.", but how to quantity the quality information could not be found in this paper. And the design and implementation of a “quality model” based on the ISO 19157:2013 standard is not clear. This paper likes a research report, then  it should be difficult for paper readers to get new knowledge from this paper. The software developed by authors for this research should be uploaded.”

The scope of the journal does not allow for more space to be devoted for  a more detailed description of the design methodology of the Quality Model based on the ISO 19157. In the design of the model, the ISO 19157 quality elements applicable to the dataset under evaluation are used. To quantify quality for each quality requirement, a quality measure is selected from the list of standardized quality measures included in Annex D of the Standard.

The combination of quality elements and quality measures selected are shown in Tables 5, 6 and 7 where the columns contain the quality elements foreseen in ISO 19157. The rows contain the entity/feature/property applied and in each cell the nameOfMeasure and the MeasureIdentification number according to Annex D of the standard are indicated. This approach is shown in more detail in Tables 8 - 11 where each row represents a cell of the table.

"Automated and semi-automatic quality inspection of spatial and descriptive information and acquisition of quantified quality information for each quality parameter of QM."

As indicated in rows 265-275, the combination of the technical assessment and the quality measure chosen to assess each quality element can be seen in tables 5, 6 and 7 of the manuscript (section QM). Orange cells indicate that the evaluation will be fully automated. Yellow and green cells indicate that the evaluation will be carried out through sampling.

Where the evaluation is carried out through sampling, the application gives users the opportunity, if they so wish, to select the sample (in the case of ISO 2859-1 selects AQL & general inspection level - code letter) and the application automatically selects the sample size using random sampling. Due to the fact that the selection of the sample is automated and the evaluation is carried out manually the process is designated as semi-automatic quality inspection.

“how to quantity the quality information could not be found in this paper”

The quantification of quality for an assessed item, results from the application of the selected quality measure. If you refer to Table 8, the quantified quality values obtained from the application of the respective measure are given in column DQ_QuantitativeResult (see column nameOfMeasure) and column ResultValueType indicates the type of value. For example where quality measure “error count” is used, the quality result is integer and in row 13 of the table the quality measure is RMSE and the quality result is in meters.

You may also refer to Row 10 of Table 8, where the application of the model on the data of the study area for Administrative units, the automated inspection of logical consistency - topological consistency with id 25 “number of invalid slivers” (Annex D, table D.26 of ISO 19157) give a quality result that the data have “3 slivers”.

Reviewer 2 Report

This article deals with a very interesting issue that concerns the formulation of a mechanism for monitoring procedures for maintaining the quality of cartographic products from cadastral data. The research work described in the article under review is scientifically complete, technically sound, supported by a large number of bibliographic references, and has a complete structure, except perhaps for one point which I will return to below.

The esteemed authors seem to have utilized a wide range and depth of knowledge on the rules and procedures of quality control, as well as an impeccable approach to geoinformation technology, through which they document all the automation processes involved in the workflow, to produce cartographic products to be presented at the end in a quality, correct and complete way, at the same time analyzing exhaustively the above-mentioned processes.

This paper could indeed be published as-is. However, the only points that put it under a partial revision are:

  1. The lack of a chapter of conclusions, which exist in the text, is simply not identified. Indeed, in my point of view, what is required to eliminate this remark is simply to add the title 'Conclusions and Future Research' between lines 562 and 563 of the text. It is very clear that what is described from line 563 to the end (line 583) of the text, is not only the conclusions drawn from the research but also the guidelines for its future approaches.
  2. A more careful application of the English language, too, would solve some - not serious - wording problems that occur sporadically throughout the text.

Best regards,

Author Response

Dear reviewer,

Thank you for your constructive comments. Please find our response in the following paragraphs: 

“This article deals with a very interesting issue that concerns the formulation of a mechanism for monitoring procedures for maintaining the quality of cartographic products from cadastral data. The research work described in the article under review is scientifically complete, technically sound, supported by a large number of bibliographic references, and has a complete structure, except perhaps for one point which I will return to below.

The esteemed authors seem to have utilized a wide range and depth of knowledge on the rules and procedures of quality control, as well as an impeccable approach to geoinformation technology, through which they document all the automation processes involved in the workflow, to produce cartographic products to be presented at the end in a quality, correct and complete way, at the same time analyzing exhaustively the above-mentioned processes.

This paper could indeed be published as-is. However, the only points that put it under a partial revision are:

  1. The lack of a chapter of conclusions, which exist in the text, is simply not identified. Indeed, in my point of view, what is required to eliminate this remark is simply to add the title 'Conclusions and Future Research' between lines 562 and 563 of the text. It is very clear that what is described from line 563 to the end (line 583) of the text, is not only the conclusions drawn from the research but also the guidelines for its future approaches.

Chapter added as suggested

  1. A more careful application of the English language, too, would solve some - not serious - wording problems that occur sporadically throughout the text.

English language was checked and appropriate modifications were made

Reviewer 3 Report

General statement

- quality and usability are two different things - it's not a matter of the different perspectives of the creator and the user

- A map is not data but a form of information (including the modeling aspect) - data quality aspects are difficult to apply to a map (large scale maps are spatial data visualizations rather than maps in the full sense of the word)

- What is noise in the context of map construction has been a fairly widely discussed topic since cartography incorporated information theory into its conceptual foundation

- The scale of 1:25000 for topographic is about the limit of where the technical aspects of conforming a map representation to a source measurement can be followed with some limitations ( there are still few aggregates, line drawing editing can be reduced to simplification, but you still can't avoid deformations for some objects)

On the other hand, I understand the motivation of the authors and I don't deny the need to document the map compilation process and to keep the basic positional constraints on the generalization for significant features

Leaving aside the conceptual problems of the topic, I find the treatment of the issue of documenting cartographic production to achieve compliance with surveying standards useful and definitely needed.

The proposed model is basically OK (it would probably be the subject of more discussion if elaborated in greater detail). The authors have obviously put together a mechanism for their proposed quality report to the database, and the discussion mentions the results within the study area - but it is not clear what the subsequent feedback will be.

There are a few comments 

- for the island-land amalgamation, it would be interesting to see the inspection record

- Tables 5 - 7 are a waste of space the information message is barely carried by a tenth of the cells - perhaps this could be done better

- Bend simplification and point removal - why is one used there and the other elsewhere - how does one actually compare the quality of these procedures (plus I understand the link to arcgis but there are a large number of similar procedures and each leads to justifiable results, how would you handle this in your quality monitoring model)

Author Response

Dear reviewer,

Thank you for your constructive comments. Please find our response in the following paragraphs: 

General statement

  • “quality and usability are two different things - it's not a matter of the different perspectives of the creator and the user”

In the introduction we generally refer to the role of quality in products derived from geospatial data such as maps, to highlight the views on quality from the perspective of producers and users. The paper is only concerned with the management and monitoring of quality in map composition rather than its usability.  

  • A map is not data but a form of information (including the modeling aspect) - data quality aspects are difficult to apply to a map (large scale maps are spatial data visualizations rather than maps in the full sense of the word)

We refer to data because maps result from the visualization of the features contained in the digital Cartographic database.

The methodology presented elaborates on the production of a map using as reference the cadastral data.  For the creation of the cartographic database this includes the features required to be portrayed on the map that will  be composed. In this context, quality monitoring refers to the transition and processing of the reference data stored in the Geospatial database and the quality of the Cartographic database resulting from their conversion and generalization. The aim of this work is to enable the cartographer to evaluate and document the quality of the cartographic database before portrayal on the map, offering evidence of control and inspection while providing more information about the final product through metadata.

Since the portrayal of the cartographic database content requires further processing such as hierarchical rendering, displacement of features etc., the application of a QM that will evaluate the quality of the representation of the features on the final map, enables the producer to further document its quality.

  • What is noise Known????? in the context of map construction has been a fairly widely discussed topic since cartography incorporated information theory into its conceptual foundation

??????

  • The scale of 1:25,000 for topographic is about the limit of where the technical aspects of conforming a map representation to a source measurement can be followed with some limitations (there are still few aggregates, line drawing editing can be reduced to simplification, but you still can't avoid deformations for some objects)

We agree in general with this comment. In the case of the application presented, the reference data used for the composition of the map are derived from a digital cadastral database. The digital cadastral database used, maintains geospatial data at 1:1,000 scale for urban areas and at 1:5,000 scale for other areas. Because of the transition from the above scales  to a scale of 1:25,000, generalization and simplification are indispensable. Furthermore, as the cadastral database has not been created for the purpose of map composition, it is necessary to transfer and process the geospatial information in two stages, from the cadastral database to the Geospatial database and from the Geospatial database to the Cartographic database. Due to of the above operations, errors may occur that affect its quality, and there is a need for acquisition of quantified quality information on specific quality measures properly documented.

On the other hand, I understand the motivation of the authors and I don't deny the need to document the map compilation process and to keep the basic positional constraints on the generalization for significant features

Leaving aside the conceptual problems of the topic, I find the treatment of the issue of documenting cartographic production to achieve compliance with surveying standards useful and definitely needed.

The proposed model is basically OK (it would probably be the subject of more discussion if elaborated in greater detail). The authors have obviously put together a mechanism for their proposed quality report to the database, and the discussion mentions the results within the study area - but it is not clear what the subsequent feedback will be.

There are a few comments

  • for the island-land amalgamation, it would be interesting to see the inspection record

Due to limitation in the extent of the paper, it is not feasible to add the inspection records for all examples.

The methodology and automation (script) of the amalgamation has been tested in several cases.

Quality assessment in this case refers to the completeness of its implementation and is described in the attached inspection record (It is part of the quality element concerning omission) that we are submitting for your consideration. An automated completeness check is carried out with reference data of the islands / islets included in the Geospatial database and based on the common entity / feature identification code between the two databases (attribute “inspireId”). The islands / islets that exist in the reference data and do not appear in the corresponding dataset of the Cartographic database, are those that were amalgamated with the coastline and through the identification code, we know exactly their position in the dataset.     

In the cases of amalgamation identified, the successful application of the methodology and the corresponding script was manually checked. If the check reveals an incorrect amalgamation, the methodology does not deliver the expected result and requires an update, re-running and re-checking of the dataset.

  • Tables 5 - 7 are a waste of space the information message is barely carried by a tenth of the cells - perhaps this could be done better

The inclusion of all the quality elements proposed by ISO 1957 in the tables intends to provide a complete picture of the quality elements selected and those not selected and which of them are applied (data or a specific property) to evaluate the feature entities in each data set.

  • Bend simplification and point removal - why is one used there and the other elsewhere - how does one actually compare the quality of these procedures (plus I understand the link to arcgis but there are a large number of similar procedures and each leads to justifiable results, how would you handle this in your quality monitoring model)

Bend simplification & point removal procedures are executed sequentially one after the other throughout the data set.

The implementation of the point removal process aims at the reduction of the data volume and the simplification of the Cartographic database. In a number of cases the length of the line segments resulting from bend simplification is so sort that it cannot be drawn by the plotter due to the resolution.

For your information an example of  the structure/content of the inspection record for LandWaterBoundary is attached.

Round 2

Reviewer 1 Report

The authors have revised this paper according to the comments. But the future research is not clear.